# The integration of tuberculosis and HIV testing on GeneXpert can substantially improve access and same-day diagnosis and benefit tuberculosis programmes: A diagnostic network optimization analysis in Zambia

Sarah Girdwood[1,2☯]*, Mayank Pandey[3☯], Trevor Machila[4], Ranjit Warrier[4], Juhi Gautam[3], Mpande Mukumbwa-Mwenechanya[4], Mariet Benade[5], Kameko Nichols[3], Lunda Shibemba[6], Joseph Mwewa[6], Judith Mzyece[6], Patrick Lungu[7], Heidi Albert[8], Brooke Nichols[1,2,4], Powell Choonga[6]

1 Department of Internal Medicine, Health Economics and Epidemiology Research Office, School of Clinical Medicine, Faculty of Health Sciences, University of the Witwatersrand, Johannesburg, South Africa,
2 Department of Medical Microbiology, Amsterdam University Medical Center, Amsterdam, The Netherlands,
3 FIND, Geneva, Switzerland, 4 Centre for Infectious Disease Research in Zambia, Lusaka, Zambia,
5 Department of Global Health, Boston University School of Public Health, Boston, MA, United States of America, 6 Ministry of Health, Lusaka, Zambia, 7 National TB and Leprosy Programme, Ministry of Health, Lusaka, Zambia, 8 FIND, Cape Town, South Africa

☯ These authors contributed equally to this work.
* sgirdwood@heroza.org

## Abstract

Diagnostic network optimization (DNO), a geospatial optimization technique, can improve access to diagnostics and reduce costs through informing policy-makers' decisions on diagnostic network changes. In Zambia, viral load (VL) testing and early infant diagnosis (EID) for HIV has been performed at centralized laboratories, whilst the TB-programme utilizes a decentralized network of GeneXpert platforms. Recently, the World Health Organization (WHO) has recommended point-of-care (POC) EID/VL to increase timely diagnosis. This analysis modelled the impact of integrating EID/VL testing for children and pregnant/breast-feeding-women (priority-HIV) with TB on GeneXpert in Zambia. Using OptiDx, we established the baseline diagnostic network using inputs for testing demand (October 2019- September 2020), referrals, testing sites, testing platforms, and costs for HIV/TB testing (transport, test, device) respectively in Zambia. Next, we integrated priority-HIV testing on GeneXpert platforms, historically only utilized by the TB-programme. 228,265 TB tests were conducted on GeneXpert devices and 167,458 (99%) of priority-HIV tests on centralized devices at baseline, of which 10% were tested onsite at the site of sample collection. With integration, the average distance travelled by priority-HIV tests decreased 10-fold (98km to 10km) and the proportion tested onsite increased (10% to 48%). 52% of EID tests are likely to be processed within the same-day from a baseline of zero. There were also benefits to the TB-programme: the average distance travelled/specimen decreased (11km to 7km),

**Data Availability Statement:** The authors do not have permission from the Ministry of Health Zambia to share the underlying data. Data are available on request from the Ministry of Health, Permanent Secretary-Technical Services, Zambia. Please contact Professor Lackson Kasonka (kasonkalm@gmail.com, ps@moh.gov.zm). The authors did not have any special access privileges that others would not have.

**Funding:** The OptiDx pilot is funded through a grant to the Foundation for Innovative and New Diagnostics (FIND) from the Bill & Melinda Gates Foundation (OPP1203377) -HA. The funder had no role in study design, data collection and analysis, decision to publish, or preparation of the manuscript.

**Competing interests:** The authors have declared that no competing interests exist.

alongside potential savings in GeneXpert device-operating costs (30%) through cost-sharing with the HIV-programme. The total cost of the combined testing programmes reduced marginally by 1% through integration/optimization. DNO can be used to strategically leverage existing capacity to achieve the WHO's recommendation regarding POC VL/EID testing. Through DNO of the Zambian network, we have shown that TB/HIV testing integration can improve the performance of the diagnostic network and increase the proportion of specimens tested closer to the patient whilst not increasing costs.

## Introduction

Since March 2021, the World Health Organization (WHO) recommends point-of-care (POC) early infant diagnosis (EID) for HIV-exposed infants and children <18months to increase same-day HIV diagnosis and conditionally recommends POC for HIV viral load (VL) testing [1]. Attention has thus turned to leveraging polyvalent diagnostic equipment that is currently in country to expand access to POC diagnostic testing or onsite testing that allows for same-day diagnosis. Understandably, tuberculosis (TB) programmes, who have invested in Cepheid's GeneXpert devices (Cepheid; Sunnyvale, CA, USA) to expand their programmes, are concerned that integration on these platforms with HIV testing might negatively affect access and performance of TB testing. However, of the 12 million TB GeneXpert cartridges procured across 140 high-burden developing countries per year in 2017 and 2018, only 1.2 tests per module per day (out of 3–4) were being run, potentially leaving additional capacity for POC HIV testing [1].

Diagnostic network optimization (DNO) can improve access to diagnostics and reduce costs through informing policy makers' decisions on diagnostic network changes [2]. DNO is a geospatial analytics approach that uses optimization techniques to model a diagnostic network and ensure the greatest access to services, whilst maximizing the overall efficiency of the system [2]. As health systems move towards providing integrated care, leveraging existing diagnostic platforms may allow for the integration of diagnostic networks and expand access to different disease diagnostics. Ensuring that current disease diagnostics are not negatively impacted is essential to efforts to integrate across programmes [2, 3]. DNO can answer questions around the impact of leveraging existing TB infrastructure to increase the proportion of EID and HIV VL testing conducted onsite.

Zambia is ranked as a high burden TB country by the WHO with an incidence of 319 per 100,000 [4]. In 2017, the Zambian National Tuberculosis and Leprosy Program adopted Xpert MTB/RIF as the first-line test for TB diagnosis to improve case detection for susceptible and drug resistant TB [5]. This precipitated a large expansion of GeneXpert devices in the country from 69 in 2016 to 291 in 2020. In comparison, VL testing and EID for HIV has predominately been performed at centralized referral laboratories resulting in long turn-around time (TAT) in some regions and impacting timely diagnosis for HIV-exposed infants as well as timely clinical action for people living with HIV (PLHIV) on antiretroviral treatment (ART). A randomized controlled trial in Zambia found that POC EID testing achieved nearly 100% same-day results, while the standard of care (SOC) took 32 days from sample collection to the guardian receiving the result, with only 41.6% of results reported back to the facility by 60 days post-randomization [6]. Typically, failure of early testing is at least in part due to centralized testing, a process that is highly vulnerable to delays, specimen loss and no result being returned to the facility or mother [7]. Recent modelling suggests that moving from the centralized SOC testing

to POC in Zambia can increase the proportion of infants initiated on ART within 60 days of sample collection from 28% to 83% (depending on the testing algorithm and platform) while being cost-effective [7, 8]. An evaluation of near POC VL implementation in seven countries in sub-Saharan Africa found that compared to conventional centralized testing, near POC VL testing (POC testing conducted onsite) improved the median time from sample collection to result return to patients from 68 to 6 days, and to clinical action from 49 days to 3 days providing compelling evidence for the adoption of near POC VL testing in countries where centralized TAT is not optimal [9].

This study forms part of a pilot study on the use of a web-based, open-access DNO tool, OptiDx [10]. The aim of OptiDx is to design diagnostic networks that enable improved access to testing of TB, HIV and other diseases and increase network efficiency. We used DNO and OptiDx to model the impact on the TB programme in Zambia of integrating priority HIV VL (VL testing for pregnant and breastfeeding women (PBFW), and children (< 15 years of age)), and EID—priority HIV testing—with TB testing on GeneXpert platforms.

## Methods

Using OptiDx software and a national cohort of retrospective, cross-sectional laboratory data, we first established the baseline diagnostic network based on 2020 (October 2019—September 2020) testing demand, referral linkages, testing sites, platforms, and costs for the HIV and TB programmes respectively. Next, as a future state scenario, we modelled the integration of priority HIV VL and EID testing on GeneXpert platforms, which have been historically only utilized by the TB programme. We then calculated the annualized device cost, variable cost/test, and sample transport cost for each scenario and disease programme, as well as access and efficiency metrics.

### Study setting and scope

All public and private health facilities (a total of 2,717) across all ten provinces in Zambia were included in the analysis. All laboratories (277) and diagnostic devices at these laboratories with HIV/TB molecular testing capability (a total of 363) were included along with 3 tests types: HIV VL, HIV EID, and Xpert MTB/RIF. HIV VL was further split into Adult VL (> 15 years of age), VL for pregnant and breastfeeding women (PBFW), and paediatric VL (< 15 years of age). Whilst we included SARS-CoV-2 and HPV testing volumes on the respective platforms, we did not optimize specifically for these test types. The location of each health facility and laboratory, as well as the available devices at each laboratory, was obtained from the Ministry of Health and implementing partners. Most specimens are referred from lower levels of care via hubs and then on to the larger molecular testing laboratories (centralized laboratories) for VL, EID, SARS-CoV-2 and HPV testing. GeneXpert devices for testing MTB/RIF are however, largely available at hubs and as a result more decentralized.

### Modelled scenarios

The historical baseline diagnostic network and scenarios were modelled using OptiDx. OptiDx was developed by the Foundation for Innovative and New Diagnostics (FIND), Llamasoft (Coupa) and the USAID's Global Health Supply Chain Program—Procurement and Supply Management to build capacity and enable countries to improve their planning of diagnostic testing services. This software tool formulates the mathematical problem using a mixed integer linear program to analyze and optimize the diagnostic supply chain network and utilizes the CPLEX algorithm to solve for the optimal solution [11, 12]. The tool uses inputs on locations of testing demand (health facilities) and testing supply (device or laboratory locations), along

with referral linkages (between health facilities, hubs and laboratories), costs and capacity/ access constraints to create a digital model of the baseline diagnostic network as well as optimize alternative network configurations. For the Zambian historical baseline, we constrained the model such that all tests are tested at laboratories as per historical testing records and referral linkages and test volumes referred between health facilities, hubs and laboratories by test type are as per the historical data. Following consultation and input from the Zambian laboratory Technical Working Group (TWG), a scenario was then created to explore a different network configuration that aligned with the TWG's objective to integrate priority HIV testing on GeneXpert devices and restrict, where possible, the model to only refer priority HIV tests to onsite testing facilities or laboratories that were close by (the integrated scenario).

## Inputs

All data was obtained from the Ministry of Health and implementing partners. All extracted aggregated laboratory data was compiled in excel, formatted and then uploaded into OptiDx to be further analysed.

## Test demand

Test demand data at a facility level represented annual test demand for the 12-month period October 2019-September 2020 (Table 1). VL and EID demand data was obtained from Disa-Lab–the laboratory information system; and TB data was obtained from the Ministry of Health Health Management Information System and triangulated with National TB Programme datasets.

## Devices and capacity

Table 2 summarises the devices available in-country for testing of VL, EID and MTB/RIF as well as the test menu considered for the different scenarios. In the integrated scenario, the Roche CAP/CTM devices were closed (as these devices will be made obsolete by the manufacturer by December 2022) and testing of priority HIV was allowed on GeneXpert devices. The 8-hour shift capacity was based on manufacturer's product information as well as discussions with the country team as to what was currently experienced at laboratories, on average, across the country. Available annual capacity per device was calculated by assuming 250 working days, 8-hour shifts, and the number of shifts available for that device at a specific laboratory. For example, larger reference laboratories primarily operated 3 shifts (24 hours), whereas small laboratories operated 1 shift a day (8 hours). Furthermore, we accommodated the test, testing demand and platforms for the HPV programme, and recent SARS-CoV-2 testing when determining the available capacity for TB and HIV testing.

**Specimen referral and access constraints.** Specimen referrals between health facilities, hubs and laboratories or directly from health facilities to laboratories along with the volumes

**Table 1. Baseline testing demand.**

| Test type | Annual baseline demand (October 2019-September 2020) |
|---|---|
| HIV VL 15+ | 892,960 |
| HIV VL PBFW | 12,237 |
| HIV VL < 15 | 53,564 |
| EID | 103,181 |
| Xpert MTB/RIF | 228,265 |
| **Total** | **1,290,207** |

**Table 2. Devices, test menu and capacity.**

| Device | Test Menu | Scenario | 8-hour shift capacity |
|---|---|---|---|
| Roche CAP/CTM 48 | EID, HIV VL <15, HIV VL 15+, HIV VL PBFW | *Baseline* | 48 |
| Roche CAP/CTM 96 | EID, HIV VL <15, HIV VL 15+, HIV VL PBFW | *Baseline* | 96 |
| Roche COBAS 4800 | HIV VL <15, HIV VL 15+, HIV VL PBFW | Baseline and Integrated | 186 |
| Roche COBAS 6800† | HIV VL <15, HIV VL 15+, HIV VL PBFW | Baseline and Integrated | 186 |
| Cepheid GeneXpert† II, IV, XVI, Infintity-48 | EID*, HIV VL <15, HIV VL PBFW | *Integrated* | 10, 20, 80, 240 |
| | MTB/RIF | Baseline and Integrated | 8, 16, 64, 192 |
| Hologic Panther† | HIV VL <15, HIV VL 15+, HIV VL PBFW | Baseline and Integrated | 320 |
| Abbott mPIMA | EID | Baseline and Integrated | 8 |

*Note. <1% of EID samples were performed on GeneXpert platforms in the baseline scenario
†Note: Performed SARS-CoV-2 and HPV testing

referred and tested on each device at a laboratory were mapped out exactly for the historical baseline. Specimens originating from multiple health facilities were aggregated at hubs (where hubs were utilized) and referred to the respective testing laboratory. For the integrated scenario, we allowed the model to optimize sample referral, however referral lanes between health facilities and laboratories, or between health facilities and hubs and then hubs and laboratories were restricted to be intra-provincial as per current practice. Additionally, we incentivized the model to choose testing laboratories that were closer to the health facility over referring further away to lower cost referral laboratories, by placing constraints on the maximum allowable distance that a specimen could be referred: 10km for priority HIV specimens and 8km for Xpert MTB/RIF (the lower distance to prioritise TB specimens over HIV as specified by the Ministry of Health). This was a soft constraint: if a testing laboratory did not exist within the radius, then specimens were referred outside of this radius for testing.

For both the historical baseline and the integrated scenario, the frequency per transportation lane was calculated as once per week if volumes at a facility were more than 50 annually, and once annually for volumes less than 50 annually. This informed the total cost for transport as described below. OptiDx calculates the distance between health facilities, hubs and laboratories using a straight-line method and adjusts for the road network using a circuity factor of 40% on straight-line distance. The speed of the transportation mode in OptiDx is fixed, and as such does not take into account seasonal variation and how this might change with the rainy season compromising road access and travel times between facilities and laboratories.

## Costs

Cost inputs in the model informed the optimization algorithm and assisted with evaluating trade-offs. Costs were sourced from implementing partners, the Ministry of Health, as well as pre-populated using international cost sources such as the Stop TB's Global Drug Facility Products Catalogue, FIND, and the Global Fund [13–15]. All costs are reported in 2020 USD and we adjusted for inflation on all costs using the inflation index from the Zambian Statistics Agency [16]. Similarly, costs that were not pre-populated from global dollar-denominated

**Table 3. Cost by device and test.**

| Device | Annual device fixed costs | Shift cost (8hrs) | Test Menu | Cost per Test |
|---|---|---|---|---|
| Roche CAP/CTM 48/96 | $1,118 | $17.62[1] | EID | $13.83 |
| | | | HIV VL | $12.50 |
| Roche COBAS 4800 | $158 | $17.62 | HIV VL | $12.50 |
| Roche COBAS 6800 | $132 | $17.62 | HIV VL | $12.50 |
| GeneXpert II | $2,415 | $11.16[2] | EID, | $18.73, |
| GeneXpert IV | $3,223 | $11.16 | HIV VL <15, | $18.91, |
| GeneXpert XVI | $10,472 | $11.16 | HIV VL PBFW, | $18.91, |
| GeneXpert Infintity-48 | $20,428 | $16.37[3] | MTB/RIF | $11.61 |
| Hologic Panther | $126 | $17.62 | HIV VL | $11.00 |
| mPIMA | $5,280 | $10.58[4] | EID | $26.34 |

Notes: Shift costs calculated based on: [1]60% of a Laboratory Technician and 40% of a Laboratory Technologists' time. [2] 20% of a Laboratory Technician, 40% Laboratory Technology, 10% Biomedical Scientist, 10% Microbiologist. [3] 80% Laboratory Technologist, 20% Biomedical Scientist.[4] 70% Laboratory Technologist.

sources were converted using the average exchange rate in the last quarter of 2020 from the Bank of Zambia. All costs are calculated from the provider-perspective. The main cost components are:

*Test costs*: This includes the supplies, consumables (including sample collection consumables) and reagents required to run a test for each device-test type combination. This cost varies with the number of tests performed.

*Device costs*: Device costs include the annual device fixed costs (cost of equipment and external quality assurance) and the annual staff cost of operating the device. The shift cost in Table 3 is the direct staff cost required to operate a device in an 8-hour day. It is calculated using estimates from the country team as to the proportion of time per day that different staff cadres spend operating a particular device. Salary data is then applied to determine the staff cost to operate a device per day. Indirect staff involved in conducting the test are not included, nor is specimen preparation time as these are often a shared activity across test types. The device fixed cost in Table 3 includes the annual equipment cost (all-inclusive procurement/ rental costs such as warranty and delivery as well as waste management) and ongoing costs related to the equipment such as training and maintenance. This cost is annualized using the estimated working life of the equipment and discounted using a discount rate of 5%. For devices that are covered by a reagent rental agreement (Hologic and Roche devices), the reagent cost is loaded (with the equipment cost component) and the equipment cost will be zero (barring the addition of any other essential ancillary equipment not included in the reagent rental agreement). The device fixed cost also includes the external quality assurance cost per device and test-type calculated as the sum of the annual panel cost for proficiency testing, the estimated number of annual tests conducted for proficiency testing multiplied by the per test cost, as well as the annual onsite visit and training cost per device. Device operating costs are allocated to test types depending on the proportion of tests conducted on the device.

*Transport costs*: A transportation cost per km for both motor vehicles (double cab and sedan) and motorbikes (the transportation modes used in Zambia) was calculated taking into account the number of vehicles and motorbikes required to operate the specimen referral system, the total annual km's driven, and costs for depreciation, fuel, maintenance, insurance and licensing, driver salaries, secondary packaging, backpacks, cooler boxes and top boxes. Total transportation costs were then estimated by first applying the cost per km ($0.60 per km for motorbikes and $0.90 per km for motor vehicles), multiplied by the distance and frequency of the respective route.

### Outcomes

For each scenario, we estimated the access metrics by test-type (priority VL, TB, EID) where access was defined in terms of the average distance travelled per specimen from specimen collection sites to testing laboratories, and the proportion of tests conducted onsite (i.e. at the site of specimen collection) on either GeneXpert devices or centralized devices. For EID tests only, we obtained data from implementing partners from all ten provinces for 3 months (January to March 2021) on the average facility to laboratory TAT (13 days) and the average intra-laboratory TAT (13 days) (the testing TAT) to obtain the total TAT of 27 days. This data was used to calculate the improvement in TAT we could expect with integration of testing on a GeneXpert device (intra-laboratory TAT of < 1 day) or onsite on GeneXpert devices (i.e. eliminating the facility to laboratory TAT). Since in the integrated scenario the Roche CAP/CTM devices were closed, EID was only conducted on GeneXpert platforms and therefore onsite testing on GeneXpert devices was considered "same-day" testing (no facility to laboratory TAT and intra-laboratory TAT of <1 day).

We also calculate average device utilization (number of tests conducted on a device over the capacity of the device—Table 2) as an indicator of efficiency. Lastly, we report total HIV/TB programme testing costs, the overall cost per TB and priority HIV test and the cost per test conducted within 10 km of a health facility.

### Sensitivity analysis

To assess the robustness of our model and outputs, we conducted a multi-one-way sensitivity analysis of the key input variables that are most likely to influence the conclusions. We calculated a change in the overall system cost, GeneXpert utilization and average distance travelled per specimen for TB and priority HIV for: (1) a change in access constraints on HIV from 10km to 40km; (2) a decrease in priority HIV test costs to the same as TB on GeneXpert; (3) a more than doubling of test demand for both the TB and HIV programmes.

### Ethics

This study was approved and granted ethical waiver by the National Health Research Authority, Zambia within the research ambit of "Laboratory Quality Improvement Research In Ministry of Health Laboratories" (NHRA000004/16/11/2021). The ethical waiver was obtained on the basis of anonymity given that solely aggregate facility-level data was used for this study.

## Results

### Location of capacity and testing demand at baseline

228,265 TB tests were conducted on 291 GeneXpert devices and 168,458 priority HIV tests (99% of EID and priority VL tests) were conducted on 44 centralized Roche/Hologic devices in 2020. Fig 1 shows baseline health facility location and testing demand (557 health facilities collecting TB specimens and 1790 health facilities collecting priority HIV test specimens) relative to the location of the centralized laboratories for conducting priority HIV testing and the more decentralized GeneXpert footprint. At baseline, overall GeneXpert utilization for TB testing was 15%, however, this varied by testing site (Interquartile Range (IQR): 5%-21%) and differed by district (IQR: 9%-24%) (Fig 2).

### Comparison of baseline and integrated scenarios

Currently, a TB specimen travels on average 11km to a testing site and priority HIV test specimens (including EID) travel 98km. With integration, the average distance travelled for priority

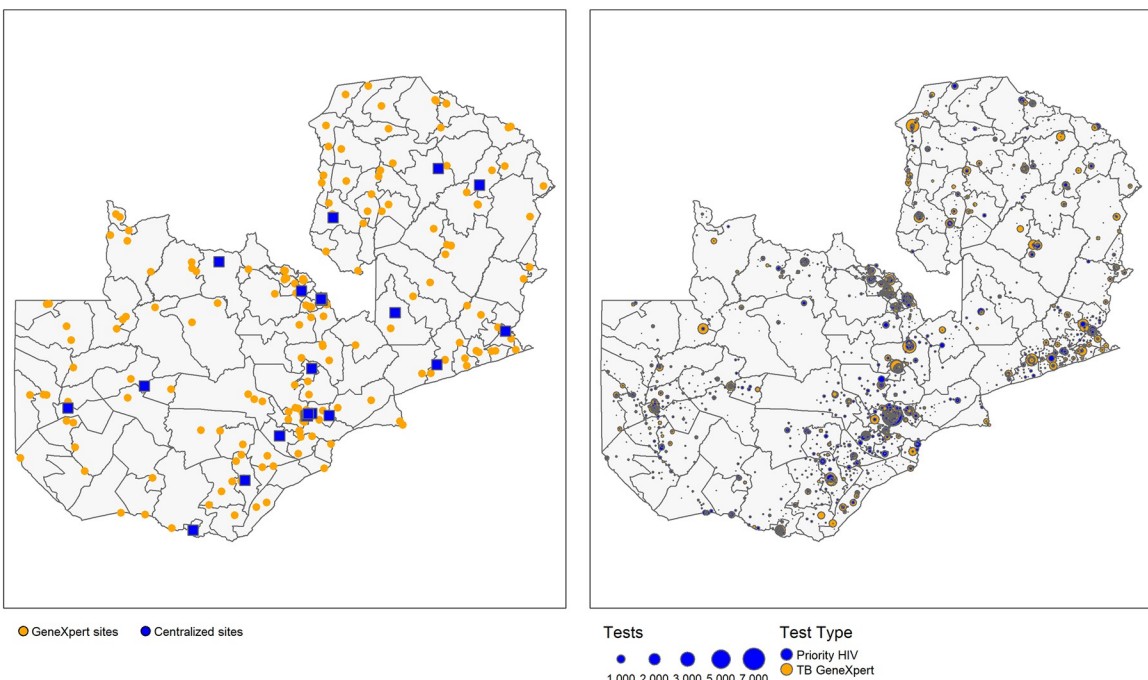

**Fig 1.** A: Baseline laboratory locations for GeneXpert devices and centralized devices (Roche, Hologic) for priority HIV viral load and EID testing, and B: Baseline testing demand locations for TB and priority HIV test specimens.

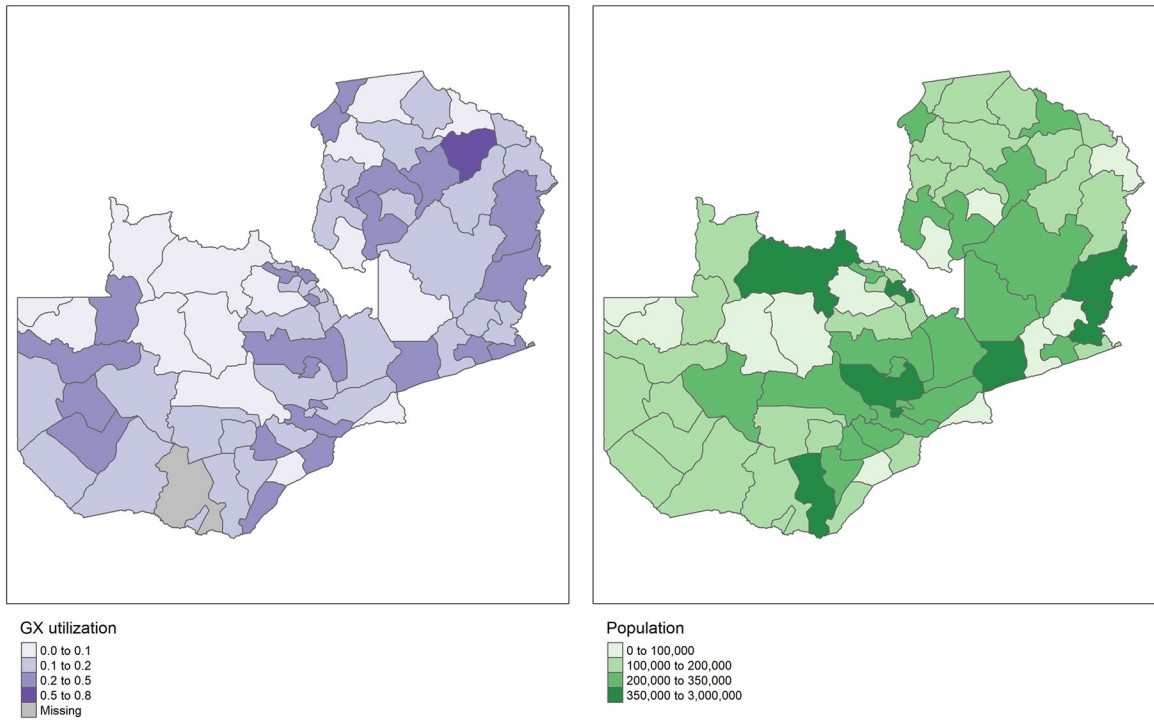

**Fig 2.** A: GeneXpert device utilization by district–Zambia, and B: Population by district–Zambia.

**Table 4. Comparison of baseline and integrated scenarios: Utilization, cost and access.**

| Scenario | | Baseline | Integrated | Difference (%) |
|---|---|---|---|---|
| Total Xpert MTB/RIF tests | | 228,265 | 228,265 | |
| Total priority HIV tests (including EID) | | 168,982 | 168,982 | (-87%) |
| (% on centralized) | | (99%) | (12%) | |
| **Overall GeneXpert utilization** | | 15% | 26% | 11% |
| Median GeneXpert utilization (IQR) | | 12% (5–21%) | 21% (8–34%) | 9% (3–13%) |
| **Average distance travelled per sample (km)** | | | | |
| | TB | 11km | 7km | -4km |
| | Priority HIV | 98km | 10km | -88km |
| **Proportion of tests performed on-site** | | | | |
| | TB | 73% | 69% | -4% |
| | Priority HIV | 10% | 48% | 38% |
| **Costs** | | | | |
| 1.Annualized device fixed cost | | | | |
| | TB | $1,318,214 | $937,430 | -$380,784 (-29%) |
| | Priority HIV | $113,436 | $764,632 | $651,196 (574%) |
| 2.Annual test cost (reagents/consumables) | | | | |
| | TB | $2,649,642 | $2,649,642 | $0 (0%) |
| | Priority HIV | $2,201,303 | $3,035,643 | $834,340 (38%) |
| 3.Annual transport cost | | | | |
| | TB | $309,452 | $148,988 | -$160,464 (-52%) |
| | Priority HIV | $1,096,396 | $701,267 | -$395,129 (-36%) |
| *Total TB testing cost* | | $4,277,308 | $3,736,060 | -$541,248 (-13%) |
| *Total Priority HIV testing cost* | | $3,411,135 | $4,501,542 | $1,090,407 (+32%) |
| *Non-priority HIV viral load testing cost* | | $12,875,243 | $12,130,217 | -$745,026 (-6%) |
| **Total cost of combined programmes (all HIV + TB)** | | $20,563,686 | $20,367,819 | -$195,867 (-1%) |
| **Total cost per test** | | | | |
| | TB | $18.74 | $16.37 | -$2.37 (-13%) |
| | Priority HIV | $20.19 | $26.64 | $6.45 (+32%) |
| **Total cost per test tested within 10km** | | | | |
| | TB | $22.92 | $19.18 | -$3.74 (-16%) |
| | Priority HIV | $69.37 | $35.55 | -$33.82 (-49%) |

HIV samples decreased 10-fold to 10km and the proportion tested onsite increased from 10% to 48%. Additionally, the average distance travelled per TB specimen reduced by 4km (from 11km to 7km) with only a marginal decrease in the proportion of tests conducted onsite from 73% to 69% (Table 4). However, of those referred offsite in the integrated scenario, a higher proportion of tests are conducted within 5km (38% vs. 20%).

The total annual cost of the combined HIV/TB testing programme (including the adult VL testing) reduced by $195,867 (1%) through integration and optimization. Importantly, there were potential savings in annualized GeneXpert device operating costs of $380,000 (30%) through cost-sharing with the HIV programme. Whilst there was a 32% increment in the overall cost per test increased for priority HIV testing from $20.19 to $26.64 due to the higher reagent and device costs for more decentralized testing on GeneXpert devices, the cost per priority HIV test conducted within 10km from a sample collection health facility decreased by 50% ($69.37 to $35.55). In the TB programme the overall cost per test and the cost per test conducted within 10km radius decreased by 13% and 16% respectively. (Table 4).

**Table 5. Results for EID and TAT.**

| Scenario | Baseline | Integrated | Difference |
|---|---|---|---|
| Total EID tests | 103,181 | 103,181 | 0 |
| Same-day tests | 0 | 53,654 | 53,654 |
| **Access** | | | |
| Proportion of EID tests performed on-site | 11% | 52% | 41% |
| On GeneXpert (same-day results) | <1% | 100% | 100% |
| Average distance travelled per EID sample | 77km | 10km | -67km |
| **Total TAT** | **26 days** | **8 days** | **-18 days** |
| Weighted average health facility to laboratory TAT | 12.5 days | 6.7 days | -6 days |
| Weighted average intra-laboratory TAT | 13 days | 1 day | -12 days |

## Access improvements for EID

Access improvements were even more stark in the case of EID testing (Table 5). 103,181 EID tests were conducted in 2020, 99% of which were on centralized platforms and <1% on mPIMA devices. Currently, 11% of EID specimens are tested onsite (either on centralized platforms or, < 1% on mPIMA devices) with an average TAT of 26 days. With integration (and the closure of the centralized Roche devices), the proportion tested onsite for EID increases almost 5-fold to 52% with decreases in the distance the sample travelled from 77km to 10km and an overall reduction in TAT by 2 and half-week. Further, 52% of EID tests are likely to be processed within the same-day from a baseline of <1%.

## Sensitivity analysis

In the multi-one-way sensitivity analysis reducing the access constraint on priority HIV tests from 10km to 40km did not significantly worsen priority HIV access outcomes suggesting that a large proportion of priority HIV tests are within 10km of a GeneXpert testing site and these devices have the capacity to conduct priority HIV testing over and above TB testing (Table 6). Decreasing the costs of priority HIV testing on GeneXpert to the same cost as Xpert MTB/RIF on GeneXpert, unsurprisingly reduced total programme testing costs relative to baseline–by 4%. Access constraints remained unchanged. With a doubling of priority HIV test demand (from 169,000 tests to 338,000 tests) and a doubling of TB GeneXpert demand (from 228,000 to 457,000) at current referring facilities (557 for TB and 1790 for priority HIV), total costs necessarily increase due to the additional reagent and consumable costs for the additional tests. GeneXpert utilization also increases but is still under 50%. Access metrics for priority HIV testing did not change, whilst access metrics decreased marginally for TB.

**Table 6. Sensitivity analysis.**

| Scenario | Integrated | (1) HIV access constraint (40km) | (2) Low Priority HIV costs on GX | (3) Doubling Demand |
|---|---|---|---|---|
| **Overall GeneXpert utilization** | 26% | 25% | 26% | 46% |
| Median device utilization (IQR) | 21% (8%-34%) | 21% (9%-37%) | 23% (10%-38%) | 45% (22%-87%) |
| **Average distance travelled per sample (km)** | | | | |
| TB | 7km | 7km | 7km | 11km |
| Priority HIV | 10km | 11km | 10km | 10km |
| **Proportion of tests performed on-site** | | | | |
| TB | 69% | 71% | 70% | 61% |
| Priority HIV | 48% | 46% | 52% | 47% |
| **Total cost** | $20,367,819 | $20,608,551 | $19,779,437 | $26,417,673 |

## Discussion

This data-focused network design model provides one of the first examples of showing the impact of integrating EID and priority HIV testing (PBFW and children) with TB testing on GeneXpert platforms in Zambia. Findings show that the use of DNO in Zambia by policy-makers and implementers to create an integrated diagnostic network for HIV and TB that leverages existing infrastructure can result in a minimal decrease in costs for HIV and TB testing programmes, whilst making substantial improvements to diagnostic access. Results show an increase in the proportion of people with access to onsite testing, the doubling of test volumes, and reduction in TAT–can all be largely achieved through the use of existing testing and specimen referral capacity with negligible cost implications.

We found substantial improvements in access metrics for priority HIV specimens (a 10-fold improvement) and 52% of EID tests are likely to be processed within the same-day from a baseline of zero allowing for a potential 2 and a half week decrease in TAT. In comparison, integration did not adversely affect access for the TB programme; moreover, the TB programme can realize potential savings in annualized GeneXpert device operating costs of $380,000 (30%) through cost-sharing with the HIV programme. Only if there is a significant increase in demand for both TB and HIV (a doubling of demand) would the TB programme be impacted in terms of onsite testing (a decrease from 73% to 61%). Whilst the overall cost per test increased by 32% for priority HIV testing due to the higher reagent and device costs for more decentralized testing, the cost per priority HIV test conducted within 10km from a sample collection site decreased by 50%. Overall, the total cost of the combined HIV/TB testing programme was reduced by 1% through integration and optimization.

The WHO guidelines that recommend POC diagnostic testing for EID and VL testing [1] also acknowledge the costs and other barriers to widespread POC adoption and, in particular, acknowledge the challenges of conducting POC VL testing for all PLHIV receiving ART due to the significant volume and testing capacity required at health facilities. It thus recommends prioritization for priority populations such as PBFW and children and adolescents as well as DNO exercises to inform POC adoption [1]. Our analysis has shown that a feasible approach to increasing the proportion of EID and priority HIV testing conducted onsite to allow for same-day and timely diagnosis is through the use of DNO to inform the optimal approach for leveraging spare capacity on GeneXpert platform within the network. This is echoed by Ndlovu et al who found that multi-disease testing on GeneXpert devices was feasible and increased access to EID, HIV VL and TB in Zimbabwe [17].

Whilst the WHO guidelines and others [18, 19] strongly promote the integration of TB/HIV testing on POC devices in tiered diagnostic networks, we are unaware of any other study that has explicitly modelled the impact of integration on the TB programme at a programmatic level. A modelling study in Kenya found integration of VL testing on GeneXpert devices (used by the TB programme) was cost-saving per HIV transmission avoided [20]. The WHO guidelines cite research that shows that POC EID testing could be cost-saving relative to the SOC per additional ART initiation when testing platforms were shared across the TB and HIV programmes [7, 8]. Whilst the de Broucker et al study on POC EID implementation in Zambia did not explicitly model TB testing volumes on GeneXpert devices, they found that integration (where excess capacity on GeneXpert devices is used for TB testing) had the largest impact on the incremental cost effectiveness ratio for the EID programme [7]. None of the studies modelled the impact from the TB programme perspective nor considered current GeneXpert device placement.

A strength of this paper is that it successfully matched facility-level testing demand to in-country testing capacity; showing the granularity of an integrated network for all HIV and TB

testing sites and the integrated referral linkages. Using sophisticated data analytics to optimize the diagnostic network, it was also able to illustrate the feasibility of a switch to GeneXpert for all priority HIV testing. This paper is very topical as it explicitly analyses the impact of this testing integration and provides comfort to TB programmes that their programmes do not need to be adversely impacted. In fact, we show that there may even be benefits from integration in terms of device cost sharing for the TB programme with the HIV programme, and optimization can allow for further access benefits for TB.

There are a number of important limitations to this paper. Firstly, we used baseline TAT data to model the impact on TAT for EID specimens if there was a reduction in health facility-to-laboratory TAT from increased onsite testing, as well as a reduction in intra-laboratory TAT through the use of GeneXpert devices. Whether these assumptions hold and whether the estimated TAT outcomes are in fact achieved when implemented need to be evaluated. For example, intra-laboratory TAT on GeneXpert devices might not be lower than larger centralized Roche equipment due to more manual processes. Secondly, whilst we have shown device operating cost savings for the TB programme, these will likely only to be realized with integrated funding structures and rely on coordination between disease programmes. Further, these device operating costs represent the replacement value of the equipment and as such do not represent immediate savings to the TB programme. Third, we calculated available capacity on GeneXpert devices taking into account manufacturer guidelines, in discussion with implementation partners. These estimates however, did not consider module breakdown time, erratic supply of cartridges, power outages, or inexperienced staff that could impact the overall capacity estimates. However, since overall utilization was low (less than 50%) we felt that our assumptions using more theoretical estimates were justified. Fourth, we used testing demand data for the period October 2019 to September 2020 and as such these figures, especially TB testing, were likely to have been impacted by the SARS-CoV-2 pandemic. TB testing decreased by 10% during the six-month period of the pandemic (April 2020-September 2020) versus the preceding 6 months; whereas HIV VL tests actually increased by 25%. Importantly, results were robust to an increase in demand for both HIV and TB testing. Lastly, this study only modelled intermediate outcomes such as access–namely, distance travelled per specimen and proportion of onsite testing performed, as well as modelled TAT for EID. It did not model the impact of these intermediate outcomes on disease outcomes–onward HIV/TB transmission and mortality. As such, the benefits of integration presented here are likely to be conservative.

## Conclusion

Through DNO of the Zambian diagnostic network, we demonstrate that the integration of HIV/TB testing on GeneXpert platforms improves the performance of the diagnostic network as measured by device utilization and a higher proportion of specimens tested closer to the patient while maintaining costs and importantly not negatively affecting the TB programme. Additionally, this analysis shows that DNO can be used to inform the integration of previously siloed programmes to increase access, decrease costs, align with WHO recommendations, and ensure sustainability of programmes by guiding decision-makers on the use of best strategies that leverage existing network capacity.

## Acknowledgments

The authors would like to thank all stakeholders from the Zambian Ministry of Health and implementing partners in Zambia who provided data and feedback on the model. The authors also wish to acknowledge Sidharth Rupani and Vinicius Brantes de Oliveira from Coupa for technical assistance with OptiDx.

## Author Contributions

**Conceptualization:** Trevor Machila, Heidi Albert, Brooke Nichols, Powell Choonga.

**Data curation:** Trevor Machila, Mpande Mukumbwa-Mwenechanya, Judith Mzyece.

**Formal analysis:** Sarah Girdwood, Mayank Pandey, Trevor Machila.

**Funding acquisition:** Heidi Albert.

**Investigation:** Mayank Pandey, Trevor Machila.

**Methodology:** Mayank Pandey.

**Project administration:** Juhi Gautam, Heidi Albert.

**Supervision:** Ranjit Warrier, Juhi Gautam, Powell Choonga.

**Visualization:** Sarah Girdwood.

**Writing – original draft:** Sarah Girdwood, Mariet Benade, Brooke Nichols.

**Writing – review & editing:** Sarah Girdwood, Mayank Pandey, Trevor Machila, Ranjit Warrier, Juhi Gautam, Mpande Mukumbwa-Mwenechanya, Mariet Benade, Kameko Nichols, Lunda Shibemba, Joseph Mwewa, Judith Mzyece, Patrick Lungu, Heidi Albert, Brooke Nichols, Powell Choonga.

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
