## [Decision Letter · Decision Letter 0]

15 Aug 2022

PGPH-D-22-00783

The integration of tuberculosis and HIV testing on GeneXpert can substantially improve access and same-day diagnosis and benefit tuberculosis programmes: a diagnostic network optimization analysis in Zambia

Dear Dr. Girdwood,

Thank you for submitting your manuscript to PLOS Global Public Health. After careful consideration, we feel that it has merit but does not fully meet PLOS Global Public Health’s publication criteria as it currently stands. Therefore, we invite you to submit a revised version of the manuscript that addresses the points raised during the review process.

Please ensure that all data underlying the findings in the manuscript are made fully available as per PLOS Global Public Health’s publication criteria. 

We look forward to receiving your revised manuscript.

Kind regards,

Elisa Lopez-Varela, MD, MPH, PhD

Academic Editor

Journal Requirements:

1. Our staff editors have determined that your manuscript is likely within the scope of our Diagnostics in Global Health Call for Papers. This editorial initiative is headed by a team of Guest Editors for PLOS GPH: Senjuti Saha (Child Health Research Foundation, Bangladesh) and Titus Divala (Public Health Scotland, University of Glasgow and University of Malawi College of Medicine). The Collection will encompass a diverse range of research articles about diagnostics in global health, including innovation and deployment of point of care diagnostics; subsets of diagnostics related to infectious diseases, chronic diseases and injuries; policies related to and regulation of diagnostics; supply chain issues; and the affordability, accessibility, and availability of essential diagnostics.  Additional information can be found on our announcement page: https://collections.plos.org/call-for-papers/diagnostics-in-global-health/

If you would like your manuscript to be considered for this collection, please let us know in your cover letter and we will ensure that your paper is treated as if you were responding to this call.  Please note that being considered for the Collection does not require additional peer review beyond the journal’s standard process and will not delay the publication of your manuscript if it is accepted by PLOS GPH. If you would prefer to remove your manuscript from collection consideration, please specify this in the cover letter.

2. Please amend your detailed online Financial Disclosure statement. This is published with the article. It must therefore be completed in full sentences and contain the exact wording you wish to be published.

b. Please expand the acronym “FIND” (as indicated in your financial disclosure) so that it states the name of your funders in full.

3. In the online submission form, you indicated that “Data is available on request from the Ministry of Health, Zambia. Please contact Powell Choonga (powellchoonga@gmail.com).”. All PLOS journals now require all data underlying the findings described in their manuscript to be freely available to other researchers, either 1. In a public repository, 2. Within the manuscript itself, or 3. Uploaded as supplementary information.

4. Please remove your figures from within your manuscript file, leaving only the individual TIFF/EPS image files.  These will be automatically included in the reviewer’s PDF.

5. Please ensure that you refer to Tables 1, 3, and 6 in your text as, if accepted, production will need these references to link the reader to the tables.

6. We do not publish any copyright or trademark symbols that usually accompany proprietary names, eg (R), (C), or TM (e.g. next to drug or reagent names). Please remove all instances of trademark/copyright symbols throughout the text, including ® (GeneXpert®) on page 23.

7. All figures and supporting information files will be published under the Creative Commons Attribution License (creativecommons.org/licenses/by/4.0/). Authors retain ownership of the copyright for their article and are responsible for third-party content used in the article. 

Figures 1 and 2: please (a) provide a direct link to the base layer of the map used and ensure this is also included in the figure legend; (b) provide a link to the terms of use / license information for the base layer. We cannot publish proprietary or copyrighted maps (e.g. Google Maps, Mapquest) and the terms of use for your map base layer must be compatible with our CC-BY 4.0 license. 

Please upload any written confirmation as an 'Other' file type. It must clarify that the copyright holder understands and agrees to the terms of the CC BY 4.0 license; general permission forms that do not specify permission to publish under the CC BY 4.0 will not be accepted. Note that uploading an email confirmation is acceptable.

Reviewers' comments:

Reviewer's Responses to Questions

**Comments to the Author**

1. Does this manuscript meet PLOS Global Public Health’s publication criteria? Is the manuscript technically sound, and do the data support the conclusions? The manuscript must describe methodologically and ethically rigorous research with conclusions that are appropriately drawn based on the data presented.

Reviewer #1: Yes

Reviewer #2: Yes

2. Has the statistical analysis been performed appropriately and rigorously?

Reviewer #1: Yes

Reviewer #2: Yes

3. Have the authors made all data underlying the findings in their manuscript fully available (please refer to the Data Availability Statement at the start of the manuscript PDF file)?

Reviewer #1: Yes

Reviewer #2: No

4. Is the manuscript presented in an intelligible fashion and written in standard English?

Reviewer #1: Yes

Reviewer #2: Yes

5. Review Comments to the Author

Reviewer #1: 1. Abstract

a. Lines 27 – 52: As a suggestion. it might be better if you put titles on abstract text. (Background or introduction, objective or aim, methods, results, conclusion, keywords.)

2. Methods

a. It is not clear what kind of study. It would be better to clarify what type of cohort study, cross-sectional, retrospective, prospective, observational, or interventional?

b. Criteria of inclusion and exclusion are no clear?

c. Sample and sampling. How was the sample size selected? It seems to us that you have collected the universe of existing data, if applicable, clarify.

d. You can clarify which variables of interest will be studied.

e. Lines 135 – 136: Data collection:

i.) not clear which system you used. Was it DisaLink, DisaTB? others? If there is any copyright, please indicate it.

ii.) Please explain more in detail how you did collect data on a database, how you anonymized them and how you were able to produce tables to be entered in your dataset analysis (i.e: Excel or…)

f. Lines 231 – 237: Analysis of statistical tests.

i.) Would it be better to clarify what kind? If you are going to do descriptive statistics in MS Excel only, or if you are going to use some advanced statistical program like SPSS, STATA, R-Project, etc.

ii.) If there is any copyright, please indicate it

3. Results:

a. The author presents four analysis tables. It would be nice if you can also divide the results into 4 subtitles followed by respective mentioned text and table, so that it is more understandable

Reviewer #2: This is an interesting study report assessing the potential impact of moving the current laboratory centralized priority-HIV [infants: early infant diagnosis (EID) and viral load (VL); women: pregnant/breastfeeding HIV/VL testing] testing to an integrated on GenXpert TB testing in Zambia through a network mathematical model. The authors set in the background a context of a country that has already managed to decentralized TB GenXpert testing, and there is a global availability of capacity to add other tests in the current TB GeneXpert cartridges.

Few issues:

1. We need a quick introduction of DNO [Diagnostic Network Optimization] somewhere in the background. This is new for a typical reader of Plos Global Public Health. This should be in the introduction/background; and I think it should also briefly be covered in the abstract.

It would be good to cover as well to cover the fundamental assumptions of DNO.

2. There is an important amount of abbreviation here. Please make a list.

3. The information about the parameters and constraints is good. The base scenario is analysed for an entire year; therefore, no intra-year variation is considered. In many parts of Africa, during the rainy season, roads become blocked, affecting logistics and access/availability to facilities and commodities. At least discuss this.

4. About the costing, please add information as to what perspective (society or the payer) these costs are considered here.

5. Abstract:

- Personally, I like to have this divided into introduction, methods, results and conclusion. So I would suggest the authors do so.

- Please add a brief statement about the DNO

6. Table 3: the row of the proportion of EID tests performed on-site. The difference between 52% and 11% is 41%. Not 36% as it is now.

6. PLOS authors have the option to publish the peer review history of their article (what does this mean?). If published, this will include your full peer review and any attached files.

**Do you want your identity to be public for this peer review?** For information about this choice, including consent withdrawal, please see our Privacy Policy.

Reviewer #1: **Yes: **Edy Nacarapa

Reviewer #2: **Yes: **Orvalho Augusto

---

## [Decision Letter · Decision Letter 1]

29 Nov 2022

The integration of tuberculosis and HIV testing on GeneXpert can substantially improve access and same-day diagnosis and benefit tuberculosis programmes: a diagnostic network optimization analysis in Zambia

PGPH-D-22-00783R1

Dear Ms Girdwood,

We are pleased to inform you that your manuscript 'The integration of tuberculosis and HIV testing on GeneXpert can substantially improve access and same-day diagnosis and benefit tuberculosis programmes: a diagnostic network optimization analysis in Zambia' has been provisionally accepted for publication in PLOS Global Public Health.

Best regards,

Julia Robinson

Executive Editor

Reviewer Comments (if any, and for reference):

Reviewer's Responses to Questions

**Comments to the Author**

1. If the authors have adequately addressed your comments raised in a previous round of review and you feel that this manuscript is now acceptable for publication, you may indicate that here to bypass the “Comments to the Author” section, enter your conflict of interest statement in the “Confidential to Editor” section, and submit your "Accept" recommendation.

Reviewer #1: All comments have been addressed

Reviewer #2: All comments have been addressed

2. Does this manuscript meet PLOS Global Public Health’s publication criteria? Is the manuscript technically sound, and do the data support the conclusions? The manuscript must describe methodologically and ethically rigorous research with conclusions that are appropriately drawn based on the data presented.

Reviewer #1: Yes

Reviewer #2: Yes

3. Has the statistical analysis been performed appropriately and rigorously?

Reviewer #1: Yes

Reviewer #2: Yes

4. Have the authors made all data underlying the findings in their manuscript fully available (please refer to the Data Availability Statement at the start of the manuscript PDF file)?

Reviewer #1: Yes

Reviewer #2: No

5. Is the manuscript presented in an intelligible fashion and written in standard English?

Reviewer #1: Yes

Reviewer #2: Yes

6. Review Comments to the Author

Reviewer #1: Revised, rearranged manuscript sounds perfect now and suitable for publication

Reviewer #2: My comments have been addressed.

7. PLOS authors have the option to publish the peer review history of their article (what does this mean?). If published, this will include your full peer review and any attached files.

**Do you want your identity to be public for this peer review?** For information about this choice, including consent withdrawal, please see our Privacy Policy.

Reviewer #1: **Yes: **Edy Nacarapa

Reviewer #2: **Yes: **Orvalho Augusto
